# Endogenous Phytohormone and Transcriptome Analysis Provided Insights into Seedling Height Growth of *Pinus yunnanensis*

Zhuangyue Lu [1,2], Qibo Wang [1,2], Zhenxin Yang [1,2], Lin Chen [1,2], Nianhui Cai [1,2] and Yulan Xu [1,2,*]

[1] Key Laboratory of National Forestry and Grassland Administration on Biodiversity Conservation in Southwest China, Southwest Forestry University, Kunming 650224, China; luzhuangyue@swfu.edu.cn (Z.L.); chengsili@swfu.edu.cn (Q.W.); yzx@swfu.edu.cn (Z.Y.); linchen@swfu.edu.cn (L.C.); cainianhui@swfu.edu.cn (N.C.)

[2] Key Laboratory for Forest Resources Conservation and Utilization in the Southwest Mountains of China, Ministry of Education, Southwest Forestry University, Kunming 650224, China

[*] Correspondence: xuyulan@swfu.edu.cn

**Abstract:** Plant height plays a crucial role in both the structure and quality of plants. *Pinus yunnanensis* is a distinctive species of the forest found in Southwest China, where the height of the plants significantly influences both yield performance and plant architecture. Although the phenotypes of *P. yunnanensis* seedlings with different plant heights were quite different at their seedling stage, the molecular mechanisms controlling the seedling differentiation remain poorly understood. This study is aimed to investigate the underlying mechanisms of *P. yunnanensis* seedling differentiation using phenotypic, transcriptomic, and endogenous phytohormone analyses. The *P. yunnanensis* seedlings were categorized into three grades, i.e., Grades A, B, and C, by mean ± 1/2 standard deviation method (H ± 1/2σ), and the seedling height and ground diameter were measured. We conducted the measurements of endogenous hormone levels in the young shoot apexes of seedlings at different grades during the fast-growth period (March). The DEGs were identified through transcriptome sequencing and analyzed by qRT-PCR validation. Significant differences were observed in the content and ratio of endogenous phytohormones among various grades of *P. yunnanensis* seedlings ($p < 0.05$). The ABA content in Grade A was prominently more than that in Grades B and C, and the order of the content of auxins was Grade B > C > A. Furthermore, when compared to Grade A, the ratios of auxins/CTKs, auxins/ABA, CTKs/ABA, and (auxins + CTKs)/ABA exhibited significant increases in Grades B and C. Moreover, GO functional annotation analysis indicated the more pronounced enrichment of DEGs in molecular functions. KEGG metabolic pathway analysis revealed notable differences in enrichment pathways between the pairwise comparisons. The "plant hormone signal transduction" pathway exhibited enrichment in the two groups, followed by "plant–pathogen interaction" pathway in the organism system that was enriched in the three groups. In addition, the results for endogenous phytohormone metabolism pathways indicate a significant up-regulation in the expression of *AUX1*, while *AHP* and *PP2C* exhibited significant down-regulation. To sum up, we aimed at investigating the underlying mechanisms of *P. yunnanensis* seedling differentiation using phenotypic, transcriptomic, and endogenous phytohormone analyses. The results suggested that individual phytohormones have a limited capacity to regulate gene expression, and seedling differentiation results from the combined regulation of multiple hormones. In addition, several candidate genes associated with phytohormone biosynthesis and signal transduction pathways were identified, including *AUX1*, *GH3*, *AHP*, *B-ARR*, *PP2C*, etc., which provided candidate genes for the following hormone-related gene overexpression and knockout experiments. These findings provide insights into the molecular genetic control of seedling height growth of *P. yunnanensis*.

**Keywords:** *Pinus yunnanensis*; seedling differentiation; plant height; endogenous phytohormones; transcriptome

## 1. Introduction

Plant height is an important aspect of plant structure and an important trait affecting plant quality [1]. Plant height, a characteristic feature of a plant species, can also respond to changes in the external environment. Variations in plant height serve as one of the most conspicuous features of growth and development [2]. With the increase in plant height, the leaves can receive more light energy, which is facilitated by photosynthesis, thus producing sufficient organic matter to provide energy for the rapid growth of the plant [3]. Meanwhile, stems mature gradually, enhancing lignification, which can reduce the phenomenon of differentiation after afforestation [4]. Studies from as early as the 1980s have demonstrated that the survival rate and the high growth of seedlings such as *Pinus staeda*, *Juglans nigra*, and *Quercus rubra* were positively correlated with the initial ground diameter during the 10 years of afforestation [5,6]. Tall seedlings perform better after afforestation under poor site conditions, and large seedlings can have advantages under the conditions of animal harm, weed competition, and serious snow pressure [7]. The study of genes associated with the ideal height of vascular plants has always been a research hotspot in the life sciences field [8,9]. Moreover, as a complex quantitative trait, plant height has complex genetic traits and unclear regulatory mechanisms [10]. Consequently, researchers selected various plant varieties to investigate aspects such as plant height regulation, phytohormone biosynthesis and signal transduction, etc. [3,11,12].

The growth rhythm of trees is the result of external and internal factors. Apart from external environmental factors affecting plant height, the expression and regulation of genes associated with plant height and growth exert significant influence on plants [1]. As signal transmission substances, plant endogenous phytohormones regulate the process of plant growth and development by promoting, inhibiting, or altering physiological activities [13]. Scientists began to study the relationship between plant height and endogenous phytohormones at the beginning of the 20th century, which proved that plant height is closely related to plant endogenous hormones [14]. Tang et al. determined the levels of endogenous auxin (IAA) and abscisic acid (ABA) in 13 *Japonica* rice varieties [15]. The results indicated that the endogenous IAA content in tall varieties was notably higher than in dwarf varieties. Phytohormones such as auxin, cytokinin (CTK), and abscisic acid play important roles in determining plant height [15]. Additionally, these phytohormones engage in interactions with ethylene, jasmonic acid, and salicylic acid throughout the developmental stages [16]. The dwarfing phenotype in plants arises from either the absence or obstruction of auxin transport function [17]. As an illustration, the overexpression of the *ZmPIN1a* (*PIN-FORMED1a*) gene in maize results in a decrease in internode length; thus, plant height is reduced due to reduced internode length [18]. Knocking out the *OsTIR1/AFB* gene induces changes in both plant height and yield in rice [19]. In alternative dwarf pine species, the dwarf phenotype is linked to endogenous phytohormones. The decrease in apical dominance in the *P. sylvestris* var. *mongolica* coincided with the notable reduction in the IAA and CTK contents contrapositive to the wild type [20]. The attributes of the dwarfed variant *P. bungeana*, characterized by stature and abundant lateral branches, could be strongly associated with the substantial rise in zeatin (ZT), the decline in IAA/ABA levels, and the increase in the ZT/IAA ratio [21]. Chen et al. demonstrated that the levels of IAA and ZT in lateral buds notably increased following the removal of apical dominance in *P. massoniana*, whereas ABA levels decreased [22]. To investigate the underlying reasons for the absence of apical dominance in *P. pygmaea*, Feng et al. determined the phytohormones content using liquid chromatography–tandem mass spectrometry. They also conducted comparative transcriptome analyses on the shoot apical meristem (SAM) and root apical meristem (RAM) of three pine species (*P. massoniana*, *P. pygmaea,* and *P. elliottii*) [12]. The results indicated that the absence of CTK and the significant accumulation of ABA- and GA-related phytohormones might contribute to the loss of shoot apical dominance and the formation of multiple branches. Additionally, the elevated expression of the GA2ox gene may be responsible for the dwarfing phenomenon [12]. These studies indicated that plant height was affected by the co-expression of multiple genes and signaling pathways.

Yunnan pine (*Pinus yunnanensis* Franch.), an evergreen arbor of the genus Pinus and subgenera Pinus (the hard pines, Diploxylon) within the Pinaceae family, is a significant tree species distributed in the subtropical mountain area of China with a warm climate [23,24]. It is an important indigenous species in the Yunnan–Guizhou Plateau and the endemic forest vegetation type in Southwest China [23–25]. This species is essential for fostering regional economic development and maintaining ecological balance [23]. *P. yunnanensis* is a pioneer tree species for afforestation on barren hills because of its light-loving, deep-rooted phenotype, drought resistance, and strong adaptability [25,26]. Currently, studies on *P. yunnanensis* seedlings are conducted more in terms of growth characteristics [27] and biomass [28]. In recent years, there have been rapid advancements in omics approaches and bioinformatics. These developments have allowed us to identify and comprehend complex regulatory networks, offering new insights into how biological systems are organized and their functioning [29,30]. For instance, Wang et al. presented the high-quality genomes of coconuts and utilized multi-omics analysis to uncover the genetic basis contributing to differences in plant height between two coconut varieties [31]. Transcriptome analysis unveiled that the candidate gene CHS regulated rapeseed plant height [32]. Zhao et al. combined transcriptomic and metabolomic analyses of plant height mutants in *Sophora davidii*. They identified differentially expressed genes (DEGs) and metabolites involved in flavonoid and phenylpropanoid biosynthesis [1]. Gan et al. utilized transcriptomic and metabolomic techniques to analyze and compare the distinctions among various spiral types of *P. yunnanensis*, and the results suggested that phytohormones (IAA, ETH) and plant–pathogen interaction significantly influence the formation and growth of spiral grains in *P. yunnanensis* [33]. Changing of the stem type of *P. yunnanensis* at the seedling stage affected the height and morphological characteristics of plants [34]. These studies have effectively identified the phenotype and molecular components closely involved in plant height. Nevertheless, due to the lack of *P. yunnanensis* genome information [35], the phenotypes of *P. yunnanensis* seedlings with different plant heights were quite different at the seedling stage. The molecular mechanisms controlling the seedling differentiation remain poorly understood. Therefore, this study aimed to reveal DEGs and associated metabolic pathways in *P. yunnanensis* plant with different heights by transcriptome and endogenous phytohormone analysis. We examined and discussed the different molecular mechanisms for the plant heights of *P. yunnanensis* based on endogenous phytohormone levels and the relationships between DEGs and different height seedlings. Furthermore, candidate key regulatory genes induced by different plant heights were screened, and quantitative reverse transcription polymerase chain reaction (qRT-PCR) analysis verified the expression of DEGs involved in phytohormone signaling. The results of this study provide insights into the molecular mechanisms underlying variations in plant height among *P. yunnanensis* seedlings. This information could serve as a basis for research focused on understanding seedling differentiation and enhancing the seedling quality of *P. yunnanensis*.

## 2. Materials and Methods

### 2.1. Plant Material

The *P. yunnanensis* seedlings used in this study were cultivated in the greenhouse of Southwest Forestry University, Kunming, Yunnan, China (25°04′00″ N, 102°45′41″ E), with normal water management. All materials were completely randomly distributed in homogenous garden experiments cultured under the same environmental conditions. The seedlings in the common garden were categorized into three grades, i.e., Grades A, B, and C, by mean $\pm 1/2$ standard deviation method ($H \pm 1/2\sigma$) [3] (H is the average height of seedlings, and $\sigma$ is the standard deviation of seedling height). The grading standard is based on the national or local seedling quality index standard [36]. Grade A seedlings: $A \geq H + 1/2\sigma$; Grade B seedlings: $H-1/2\sigma \leq B < H + 1/2\sigma$; and Grade C seedlings: $C < H-1/2\sigma$. Shoot apexes of the initially graded *P. yunnanensis* were selected for sample collection at approximately 8:00 a.m. in the early mornings during the middle of January,

March, May, August, and December 2022, with a total of five stages (Table S1). By analyzing and comparing the results of the measured data (plant height, ground diameter, etc.), *P. yunnanensis* shoot apexes were collected from A, B, and C plants at the fast-growing stage (March 2022) with three biological replicates. Each grade consisted of randomly collected from 9 *P. yunnanensis* shoot apexes, with a total of 27 seedlings. All experiments were conducted in the common garden conditions, and no treatment was performed except for normal watering and weeding. Because of the small number of gradation changes between the seedlings, these seedlings were not selected for collection. Following harvesting, all the tissue samples were promptly frozen in liquid nitrogen and preserved at −80 °C for subsequent RNA extraction and endogenous phytohormone analysis.

*2.2. Transcriptome Sequencing and Bioinformatics Analysis*

The collected tissue samples were provided to PANOMIX Biomedical Technology Co., Ltd. (Suzhou, China) for RNA-seq. Total RNA was isolated utilizing a magnetic tissue total kit (DP762, Tiangen, Beijing, China). PANOMIX Biomedical Technology Co., Ltd. (Suzhou, China) performed the Illumina RNA-Seq. A nanophotometer spectrophotometer (IMPLEN) and an Agilent Bioanalyzer 2100 system were used to assess the RNAs' purity and integrity, respectively. Poly(A) mRNA was enriched using magnetic beads containing oligo (dT), followed by the random fragmentation of the mRNA into fragments ranging from 200 to 300 bp. First-strand cDNA (RNase H-) was synthesized using M-MuLV reverse transcriptase and random hexamer primers. Next, the synthesis of the second cDNA strand was conducted using DNA Polymerase I and RNase H. Exonuclease/polymerase activities were employed to transform the remaining overhangs into blunt ends. Double-stranded cDNAs were ligated with sequencing adapters. After amplification and purification, cDNA libraries were created and sequenced using the Illumina HiSeqTM 2000 system platform.

Raw reads were purified by discarding reads containing only adapter and low-quality reads (q) < 20 from raw data. The GC content of the clean reads was computed, and the base quality was evaluated by generating Q20 and Q30 values using FastQC. Subsequently, Trinity 2.5.1 was employed to assemble the high-quality reads into the unigene sequence [37]. The reference genome used Pita.2_01.fa (*Pinus taeda*), and the genome version was v2.01 (https://treegenesdb.org/FTP/Genomes/Pita/v2.01/, accessed on long-term). The filtered reads were mapped to the reference genome utilizing HISAT2 v2.0.5. The quantification of gene expression levels were calculated as follows:

$$\text{FPKM} = \frac{\text{RCg (Number of reads mapped to the gene)} \times 10^9}{\text{RCpc (Number of reads mapped to all proteincoding genes)} \times \text{L (Length of the gene in base pairs)}}$$

Anders and Huber [38] published a report that served as a reference for identifying changes in gene expression. The selected high-quality reads underwent normalization by the DESeq package in R. The abundance change in each transcript was quantified using the raw counts data and expressed as log2 (fold change). Differentially expressed genes were those that had |log2 fold change| > 1 and $p$-value < 0.05 [39]. All DEGs underwent gene ontology (GO) term annotation (http://www.geneontology.org/, accessed on 27 February 2024) [40] studies for functional analysis. Using the top GO to perform GO enrichment analysis on the differential genes, $p$ value was calculated by hypergeometric distribution method ($p < 0.05$). Using the Kyoto Encyclopedia of Genes and Genomes (KEGG) (http://www.genome.jp/kegg/, accessed on 27 February 2024) [41], the identified functional genes were mapped, which is useful for the further analyses of the networks of the genome [42]. Transcription factor prediction entails comparing plants with databases like PlantTFDB (Plant Transcription Factor Database) to forecast transcription factors and their corresponding families. This is followed by an examination of mRNA transcription levels.

### 2.3. Determination of the Contents of Endogenous Phytohormones

The young shoot apexes of A, B, and C seedling grades (approximately 0.5 g fresh weight per sample) were collected. All the tissue samples were promptly frozen in liquid nitrogen and preserved at $-80\,^{\circ}\text{C}$. Each of the three grades was subjected to three biological replicates. The levels of endogenous phytohormones were assessed by PANOMIX Biomedical Technology Co., Ltd. (Suzhou, China). Phytohormone quantification was conducted using ultra-performance liquid chromatography coupled with tandem quadrupole mass spectrometry, which included an electrospray interface (UPLC-MS/MS). The raw data were collected utilizing Analyst 1.7.2 software, and the results were qualitatively and quantitatively analyzed by the SCIEX OS V2.0.1.48692 software. The concentration in samples was calculated using the formula that incorporates the peak area ratio of target compounds to an internal standard [43].

### 2.4. Validation of Transcriptome Data Using qRT-PCR

The results of RNA-seq analysis were verified using qRT-PCR analysis. Total RNA was reverse-transcribed using QIAGEN QuantiNova Rev. Transcription Kit (QIAGEN, Hilden, Germany) following the manufacturer's instructions. QRT-PCR was conducted using PowerUp SYBR Green Master Mix Kit on ABI QuantStudio6 Real-Time PCR System (Applied Biosystems, Carlsbad, CA, USA). Primers specific to the identified candidate genes were designed using Primer 3.0. The following thermal cycling parameters were used, i.e., $95\,^{\circ}\text{C}$ for 2 min, followed by 40 cycles of $95\,^{\circ}\text{C}$ for 10 s and $60\,^{\circ}\text{C}$ for 10 s, and $72\,^{\circ}\text{C}$ for 40 s in a 10 µL volume. Each reaction included three biological replicates, with the gene (PITA_43709) used as an internal reference. The relative expression level of each gene in seedling Grades A, B, and C was calculated by the $2^{-\Delta\Delta Ct}$ method [44].

### 2.5. Data Statistics and Analysis

Data analysis was performed using the SPSS 25.0 software (SPSS Inc. version 25.0, IBM, Chicago, IL, USA, 2017) with the Duncan's test. The contents of endogenous phytohormones in three seedling grades were analyzed by single-factor analysis of variance (ANOVA) and Pearson correlation analysis. The Logistics equation of time to plant height increment was fitted with DPS data processing system for seedling growth curve. Here, the growth curves correlate the seedling height to the cultivated time according to Logistics equation, $Y = K/(1 + e^{a-bt})$ [45], and the components of the equation are as follows:

Y is the plant height at different times;

K is the growth limit of the seedling height;

t is the cultivated time;

a and b are the parameters of the Logistics model.

Therefore, the beginning of a fast-growing period is described as $t_1 = (a - 1.317)b^{-1}$, the end of a fast-growing period as $t_2 = (a + 1.317)b^{-1}$, and the duration of a fast-growing period as $t_s = ab^{-1}$.

The R4.2.1 software, Ropls [39], was used for Pearson correlation and principal component analysis (PCA). The Kruskal–Wallis test (for three and more groups) and the Mann–Whitney–Wilcoxo test (for two groups) were performed to investigate the significant differences among groups ($p$ value $< 0.05$). The results from qRT-PCR and the measurements of phytohormone levels were graphed using Origin 2021 and Excel 2021 software. The FPKM values obtained from RNA-seq were compared with the relative expression levels normalized from the qRT-PCR results. SPSS 25.0 was used to analyze the data by ANOVA, and Duncan's method was used for pairwise comparison and significant difference analysis.

### 3. Results

#### 3.1. Plant Height Differences of P. yunnanensis among Different Grades

Logistics model could well reflect the growth rhythm of *P. yunnanensis* seedlings and provide a reliable quantitative model for the analysis and prediction of different growth stages of *P. yunnanensis*. According to the growth of seedling height in different months, the Logistics growth curve of seedling height was calculated and fitted (Figure 1, Table S1). As shown in the figure, the height growth of different grades of seedlings showed an "S" curve with time, that is, the growth rhythm of "slow–fast–slow". At the same time, both the seedling height and ground diameter of *P. yunnanensis* showed the order of Grade A > B > C, and the increment in the seedling height of the three grades of *P. yunnanensis* in March was prominently more than that in other months (Table S1), when seedlings were entering the fast-growing period (March) (Figure 1). Therefore, the beginning of a fast-growing period $t_1$ was at 69~80 d, the end of a fast-growing period $t_2$ was at 211~237 d, and the duration of a fast-growing period $t_s$ was at 140~159 d (Table 1). After entering the fast-growing period of seedling height, the growth rates of seedling height of different grades were significantly different. The height growth rate of Grade A seedlings was remarkably higher than that of Grade B and C seedlings, and the overall growth rhythm of the three grades of seedlings was relatively consistent. All showed a slow–fast–slow S-type growth curve. Therefore, the length of seedling growth period does not primarily account for the variation in seedling growth. Instead, the timing and duration of the fast-growing period, whether it occurs sooner or later, directly impact seedling growth.

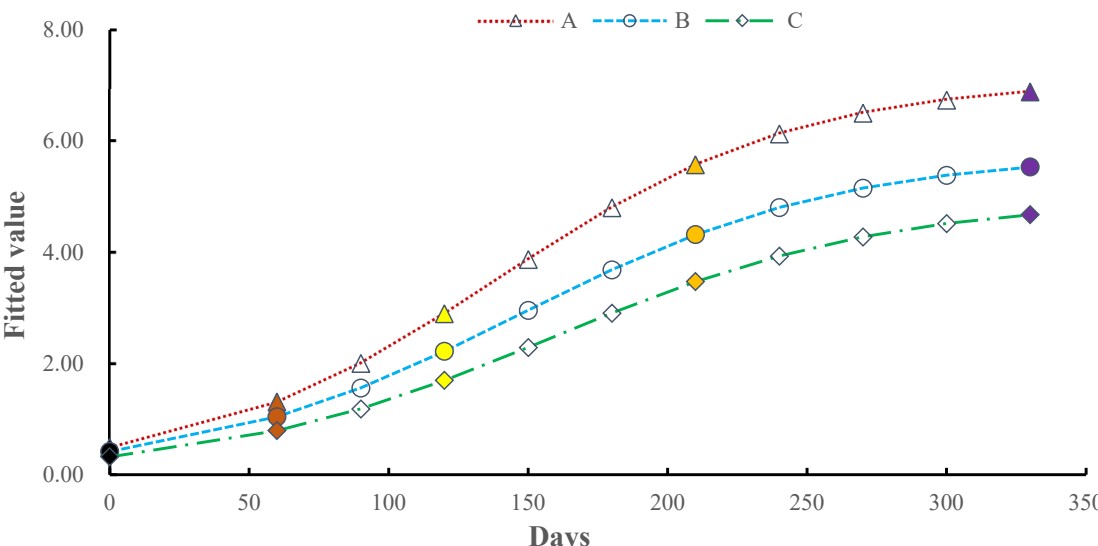

**Figure 1.** Plant height growth curves of different grades of *P. yunnanensis* seedlings in different months. Black circle: January (0 days in the figure refers to the time of the first seedling height measurement); red: March; yellow: May; orange: August; and purple: December; white: the month in which no samples were collected, including April, June, July, September, October, November.

**Table 1.** Logistics model fitting of plant height increment in different grades of *P. yunnanensis*.

| Grade | k | a | b | $t_1$ | $t_2$ | $t_s$ | $R^2$ | F Value | *p* Value |
|-------|-----|-----|-----|-----|-----|-----|-----|-----|-----|
| A | 7.0824 | 2.6044 | 0.0186 | 69 | 211 | 140 | 0.9938 | 645.076 | 0.0001 |
| B | 5.763 | 2.5539 | 0.0174 | 71 | 222 | 147 | 0.9924 | 523.766 | 0.0001 |
| C | 4.9408 | 2.6628 | 0.0168 | 80 | 237 | 159 | 0.9923 | 518.109 | 0.0001 |

### 3.2. Analysis of Endogenous Phytohormone Content in P. yunnanensis Seedlings of Different Grades

To investigate the difference in endogenous hormones in *P. yunnanensis* seedlings, the levels of 15 endogenous phytohormones, including ethylene (ACC); abscisic acid (ABA); salicylates: salicylic acid (SA) and glycosylated salicylic acid (SAG); auxins: indole-3-acetic acid (IAA), indole-3-formaldehyde (ICAld), and indole-3-methyl acetate (Me-IAA); jasmonic acids: jasmonic acid (JA), jasmonic acid-isoleucine (JA-lle), and 12-oxadienoic acid (OPDA); cytokinins: isopentenyl adenine nucleoside (IPA), isopentenyl adenine (IP), trans zeatin nucleoside (TZR), trans zeatin (TZ), and dihydrozeatin (DZ), were measured in the three seedling grades during their rapid growth period (Figure 2).

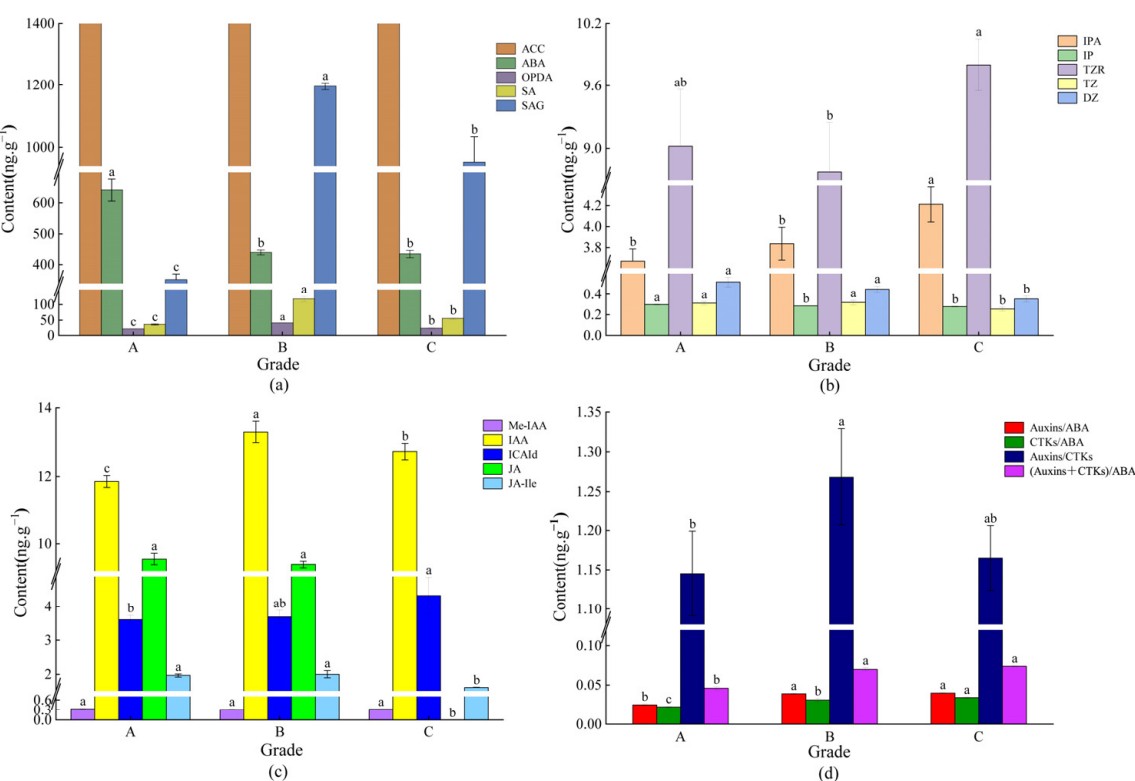

**Figure 2.** Endogenous phytohormone content of *P. yunnanensis* seedlings of different grades. Error bars represent the standard error of the mean (SD) for each data point, with a sample size of three (n = 3). (**a**) ACC (1-amino-1-cyclopropanecarboxylic acid), ABA (abscisic acid), OPDA (12-oxadienoic acid), SA (salicylic acid), and SAG (glycosylated salicylic acid). (**b**) IPA (isopentenyl adenine nucleoside), IP (isopentenyl adenine), TZR (trans zeatin nucleoside), TZ (trans zeatin), and DZ (dihydrozeatin). (**c**) Me-IAA (indole-3-methyl acetate), IAA (indole-3-acetic acid), ICAd (indole-3-formaldehyde), JA (jasmonic acid), and JAIle (jasmonic acid-isoleucine). (**d**) Auxins/ABA, CTKs/ABA, Auxins/CTKs, (Auxins + CTKs)/ABA. Because the ACC content value is too large, it cannot be marked with significance. Lowercase letters denote significant differences among different grades at a significance level of $p < 0.05$, as determined by Duncan's test. Different colors indicate different hormones.

Significant differences were observed in the content and ratio of phytohormones among various grades of *P. yunnanensis* seedlings (Figure 2, *p* < 0.05). Based on the phytohormonal data, compared to the Grade A seedlings, the ABA content displayed significant declines in Grade B and C seedlings, and the other phytohormones (OPDA, SA, SAG) were significantly increased in Grade B and C seedlings (Figure 2a). Most kinds of cytokinins were detected, in which the content of TZR was the highest, and the order of TZR was Grade C > A = B. Compared with Grade C seedlings, except for the hormones of TZR and IPA, the phytohormones of IP (A: 0.297, C: 0.278), TZ (A: 0.313, C: 0.254), and DZ (A: 0.518, C: 0.353) were higher in Grade A seedlings (Figure 2b). IAA is the most prevalent natural plant auxin. The level of IAA in Grade B seedlings was more than that in Grade C and A seedlings. Relative to the Grade C seedling, the ICAld content of Grade A seedlings decreased, while the content of JA-lle increased significantly. JA was only detected in Grade A and B seedlings, and there was no significant difference in Me-IAA (Figure 2c). Auxins and CTKs are growth-promoting factors, while ABA is a growth-inhibitory factor. The proportional relationship between growth-promoting factors and growth-inhibitory factors affects growth and development. Auxins/ABA and (auxins + CTKs)/ABA of Grade B and C seedlings were significantly higher than those of Grade A seedlings. The CTKs/ABA ratio rank was C > B > A. At the same time, auxins/CTKs had the highest ratio in Grade B seedlings (Figure 2d). In general, the contents of ABA and ETH were the highest in Grade A seedlings, JA and SA were the highest in Grade B seedlings, and auxins and CKs were the highest in Grade C seedlings. The difference in hormone contents led to the difference in seedling morphology. The ratios of auxins/CTKs, auxins/ABA, CTKs/ABA, and (auxins + CTKs)/ABA were significantly increased in Grade B and C seedlings compared with those in Grade A seedlings. The high growth of *P. yunnanensis* may be regulated by various phytohormones, and the levels of phytohormone ratios regulate the different plant heights.

*3.3. Transcriptome Analysis of P. yunnanensis*

The different plant height samples of *P. yunnanensis* were subjected to transcriptome sequencing. Transcriptome sequencing yielded raw reads ranging from 38,404,410 to 43,968,286. After filtering out the joint and the low-quality reads (excluding reads with a removal ratio of N of more than 10%), the total clean sequence data ranged from 36,150,428 to 41,333,726. Q20 > 97.92%, and Q30 > 93.86%. The average GC content across the nine libraries was 46%, with an average of 90.49% of reads being uniquely mapped (Table 2). Briefly, this suggests that the sequences were of high quality and could be used for further investigation. Pearson correlation analysis revealed strong correlations among the three replicates of each sample (Figure 3a). Principal component analysis (PCA) grouped the samples into three clusters, each corresponding to one of the three grades of *P. yunnanensis* seedlings. The first and second principal components, which represented 87% of the total variance (PC1 = 61%, PC2 = 26%), corresponded to the three grades of seedling growth and development (Figure 3b). Both PCA and correlation analysis suggested that the samples were of high quality and appropriate for further assessment.

**Table 2.** Quality control and comparison of *P. yunnanensis* transcriptome sequencing.

| Sample | Raw Reads | Clean Reads | Clean Bases | Q20 (%) | Q30 (%) | GC Content (%) | Total Mapped Reads (%) | Uniquely Mapped Reads (%) | Multiple Mapped Reads (%) |
|---|---|---|---|---|---|---|---|---|---|
| A1 | 42,460,252 | 39,989,758 | 6,038,453,458 | 97.99 | 93.99 | 46.27 | 26,778,583 (66.96%) | 24,251,156 (90.56%) | 2,527,427 (9.44%) |
| A2 | 38,404,410 | 36,150,428 | 5,458,714,628 | 98.23 | 94.63 | 46.19 | 24,144,232 (66.79%) | 21,829,450 (90.41%) | 2,314,782 (9.59%) |
| A3 | 40,235,024 | 37,840,278 | 5,713,881,978 | 98.07 | 94.19 | 46.21 | 25,274,245 (66.79%) | 22,892,443 (90.58%) | 2,381,802 (9.42%) |
| B1 | 40,302,140 | 37,661,768 | 5,686,926,968 | 98.09 | 94.3 | 46.28 | 25,156,207 (66.80%) | 22,701,899 (90.24%) | 2,454,308 (9.76%) |
| B2 | 42,717,490 | 40,137,824 | 6,060,811,424 | 98.09 | 94.27 | 46.29 | 26,847,872 (66.89%) | 24,265,389 (90.38%) | 2,582,483 (9.62%) |
| B3 | 43,968,286 | 41,333,726 | 6,241,392,626 | 98.13 | 94.36 | 45.99 | 27,572,585 (66.71%) | 24,950,400 (90.49%) | 2,622,185 (9.51%) |
| C1 | 41,772,378 | 39,178,750 | 5,915,991,250 | 98.26 | 94.71 | 46.19 | 26,302,198 (67.13%) | 23,829,061 (90.60%) | 2,473,137 (9.40%) |
| C2 | 40,706,310 | 38,272,706 | 5,779,178,606 | 98.19 | 94.54 | 46.10 | 25,460,467 (66.52%) | 23,050,168 (90.53%) | 2,410,299 (9.47%) |
| C3 | 39,505,860 | 37,142,862 | 5,608,572,162 | 97.92 | 93.86 | 46.08 | 24,540,701 (66.07%) | 22,234,552 (90.60%) | 2,306,149 (9.40%) |

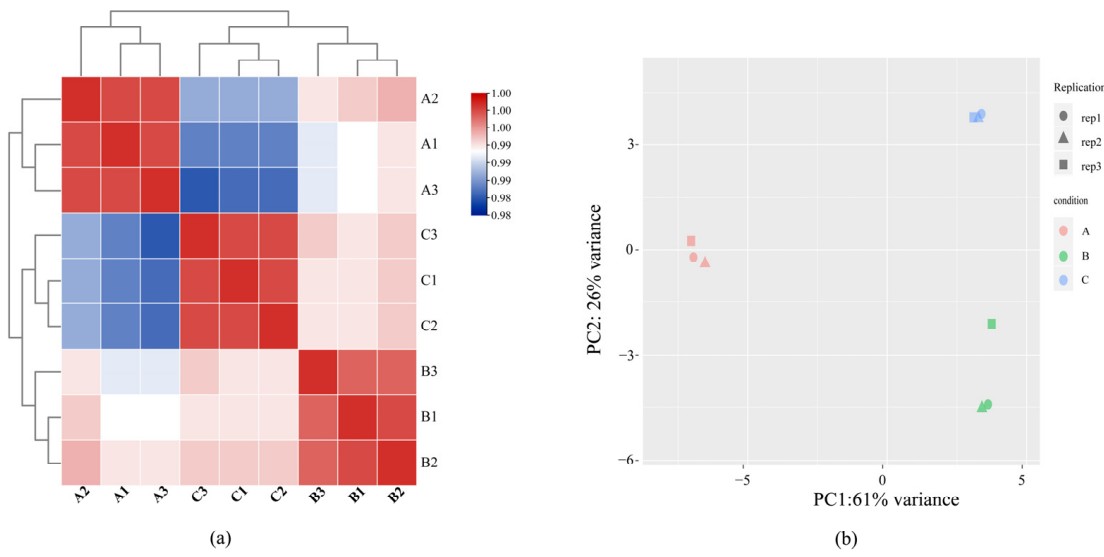

(a)  (b)

**Figure 3.** The pairwise Pearson correlation coefficients were calculated for the sequencing data obtained from three replicates of each sample collected at the three levels. (**a**) Principal component analysis (PCA) was performed on the transcriptome data collected from samples at the three levels (**b**).

### 3.4. Differentially Expressed Genes among Different P. yunnanensis Seedling Grades

Transcriptome data from *P. yunnanensis* seedlings at the three different grades were compared pairwise. A total of 1338 DEGs were identified, with the most DEGs (632) observed in A-vs-B and A-vs-C, and the fewest (576) seen in B-vs-C. Additionally, 24 genes revealed notably differential expression among the three grades (Figure 4a). In A-vs-B, 394 genes were up-regulated, and 238 genes were down-regulated. In A-vs-C, 277 genes were up-regulated, and 355 genes were down-regulated. However, the lowest numbers of DEGs were seen in B-vs-C, with 195 and 381 genes being up-regulated and down-regulated, respectively (Figure 4b). We explored and discovered the biological relationship between genes through the cluster analysis of differential gene expression. Cluster analysis revealed

that the clustering regions of DEGs with high or low expression were different among different groups of samples, which led to different clustering results. However, they existed in cross-sectional regions. The expression patterns of the same group of samples were similar (Figure 4c).

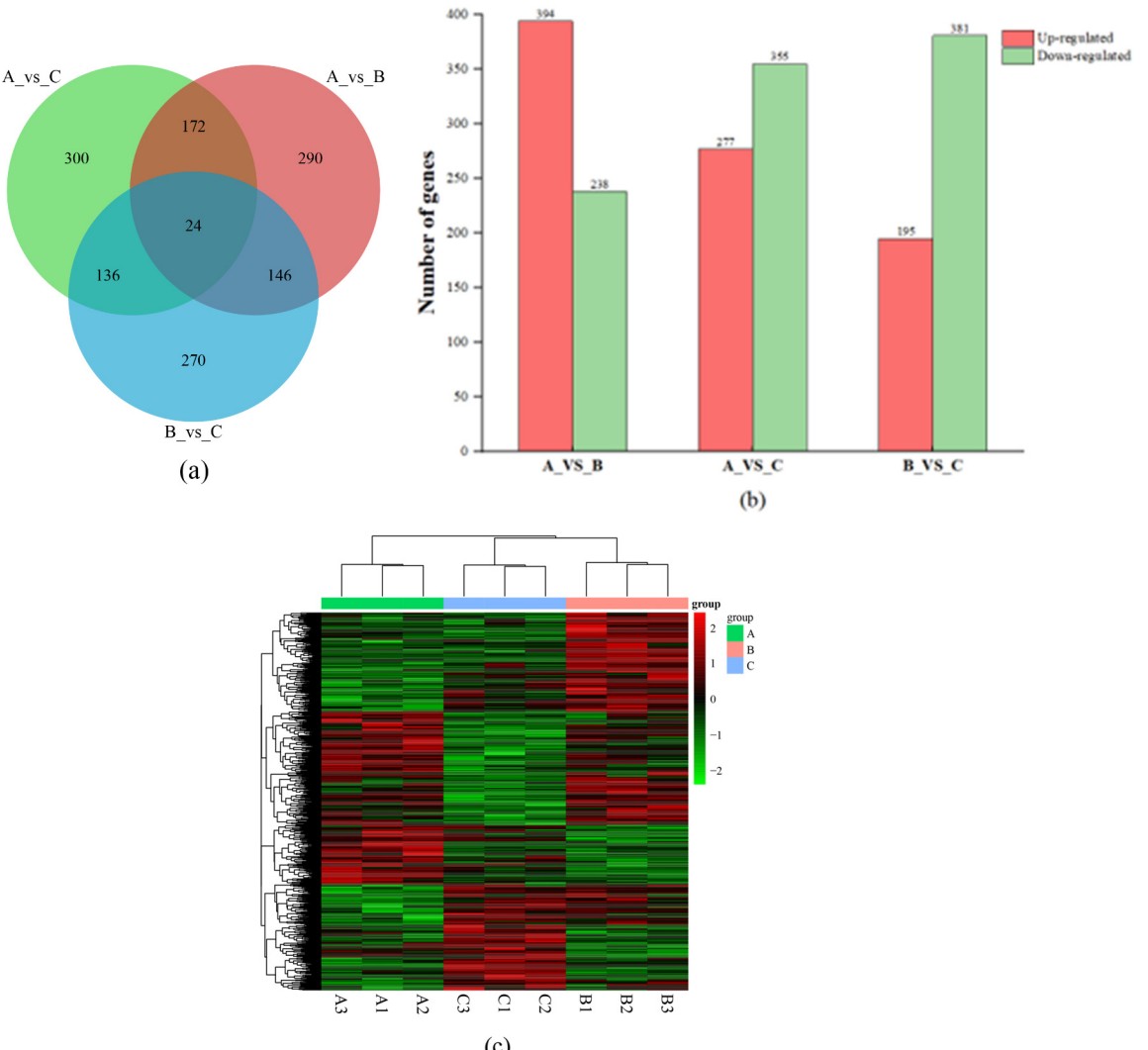

(a)

(b)

(c)

**Figure 4.** Pairings of different grades (A, B, C) of *P. yunnanensis* seedlings were compared to identify the characterization of DEGs. (**a**) Venn diagram of three pairs comparing DEGs numbers (A-vs-B, A-vs-C, and B-vs-C). (**b**) The number of upper and lower DEGs in each pair is compared. Green represents the down-regulation of DEGs, while red represents the up-regulation of DEGs. (**c**) Clustering analysis of DEGs. Horizontal represents genes, vertical represents samples, red represents high-expression genes, and green represents low-expression genes.

### 3.5. Functional Annotation by GO and Enrichment Analysis of KEGG Metabolic Pathway of Differentially Expressed Genes

3.5.1. GO Functional Annotation Enrichment Analysis

DEGs identified from three different groups, with pairwise comparisons of the most DEGs were annotated against the GO databases. We analyzed the cellular component (CC), molecular function (MF), and biological process (BP) of differentially expressed genes (Figure 5a–c). In total, 122,490 unigenes were identified and categorized into different functional sub-groups across the three principal GO categories (CC, MF, and BP). In A-vs-B, the most enriched GO terms were "extracellular region" in cellular component; "catalytic activity" and "antioxidant activity" in molecular function; and "immune system process"

and "multi-organism process" in biological process (Figure 5a). In A-vs-C, these terms included "membrane" in cellular component; "catalytic activity" in molecular function; and "cell killing" in biological process (Figure 5b). In B-vs-C, these terms included "membrane" in cellular component; "catalytic activity" and "antioxidant activity" in molecular function; and "immune system process", "metabolic process", and "response to stimulus" in biological process (Figure 5c). The "catalytic activity" pathway was significantly enriched in the three groups. The term indicated collectively the proteins involved in enzyme activities. It may be referred to as a pathway associated with the growth of *P. yunnanensis* seedlings. The findings showed that there was significant gene enrichment in cell composition, molecular function, and biological process, which indicated notable differences in cell, molecular function, and biology during the growth and development of different grades of *P. yunnanensis* seedlings. It is suggested that the genes enriched in these functional categories may be involved in plant growth regulation.

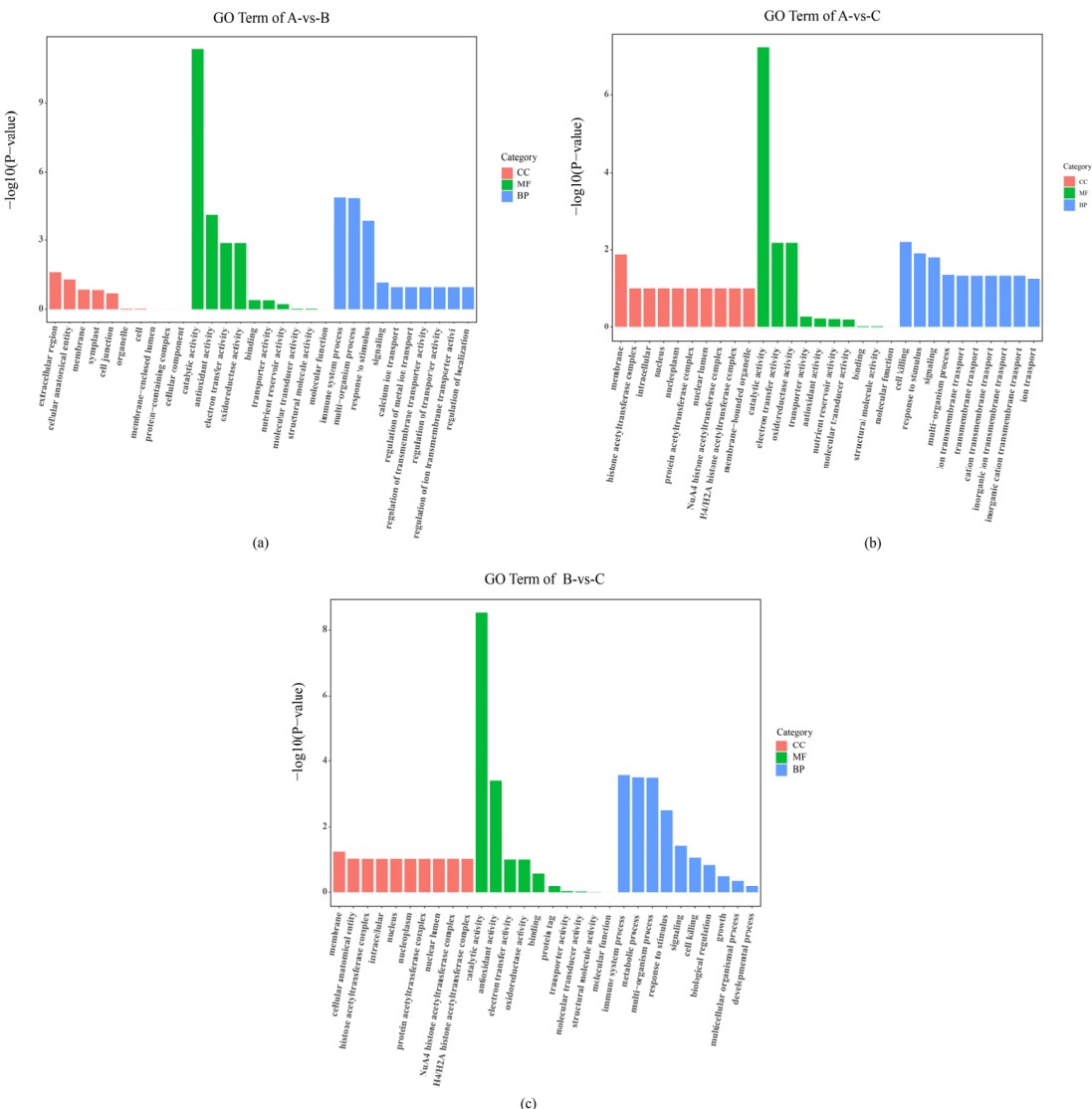

**Figure 5.** GO enrichment analysis of DEGs. The abscissa is the GO term of enrichment, and the ordinate is the *p*-value of the term enrichment. (**a**) GO term of A-vs-B; (**b**) GO term of A-vs-C; and (**c**) GO term of B-vs-C.

### 3.5.2. KEGG Metabolic Pathway Enrichment Analysis

KEGG functional annotation analysis was performed on the differentially expressed genes. In A-vs-B, the KEGG enrichment analysis classified the DEGs into three major cate-

gories. The top pathways included "plant hormone signal transduction" in environmental information processing, "linoleic acid metabolism" in metabolism, and "plant–pathogen interaction" in organismal systems (Figure 6a,b). In A-vs-C, the top pathways included "MAPK signaling pathway plant", "plant hormone signal transduction", "flavonoid biosynthesis", and "plant–pathogen interaction" (Figure 6c,d). In B-vs-C, these DEGs were classified into five major categories, "endocytosis" in cellular processes, "phosphatidylinositol signaling system" in environmental information processing, "non-homologous end joining" in genetic information processing, "linoleic acid metabolism" in metabolism, and "plant–pathogen interaction" in organismal systems (Figure 6e,f). There were significant differences in enrichment pathways between the pairwise comparisons. The "plant hormone signal transduction" pathway exhibited enrichment in the two groups, followed by "plant–pathogen interaction" pathway in the organism system was enriched in the three groups. Different pathways appear to be important at different plant heights. These results indicated that the above pathways could be closely associated with the plant height of *P. yunnanensis* and affect the growth of seedlings.

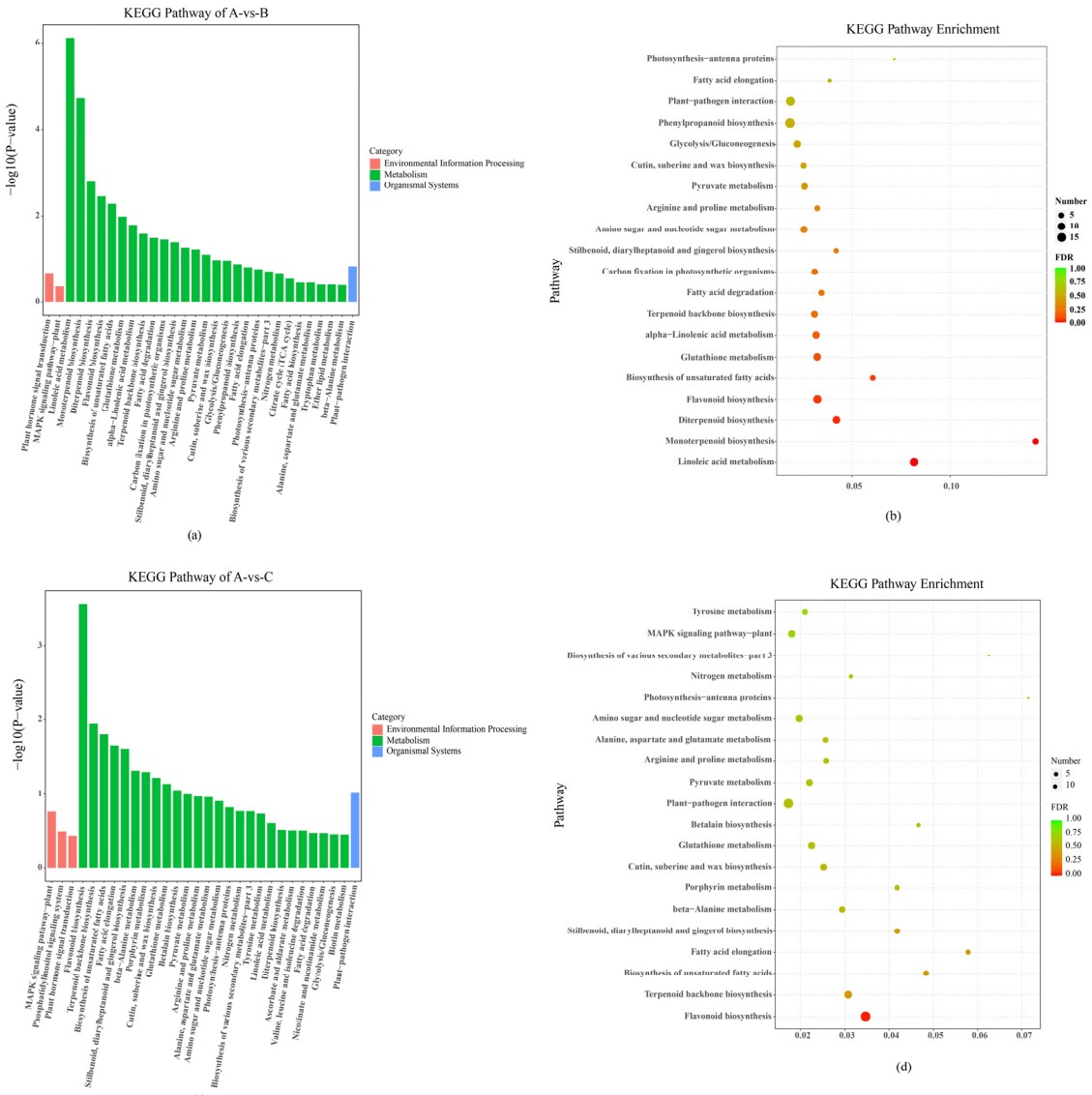

**Figure 6.** *Cont.*

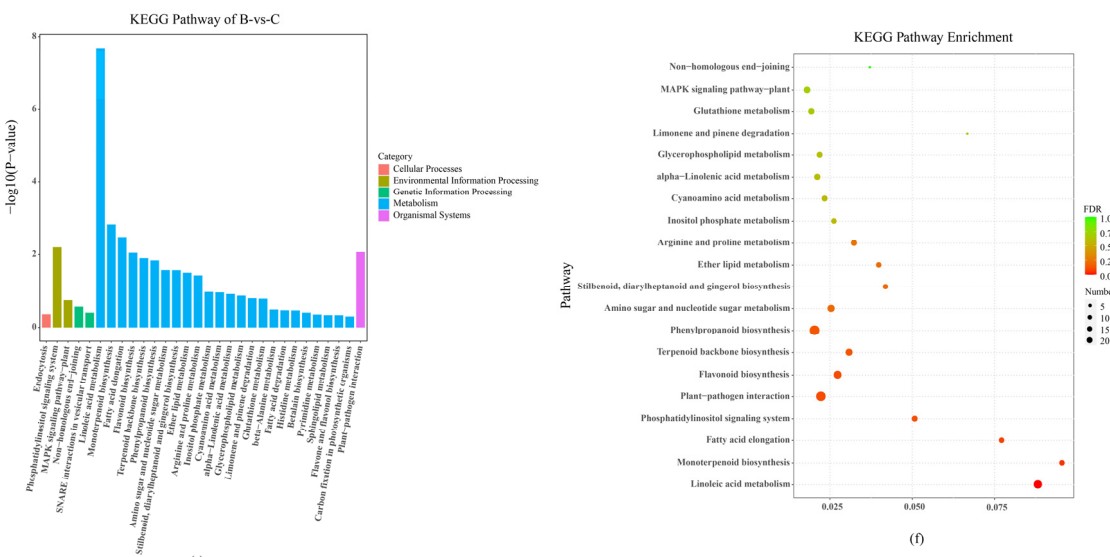

**Figure 6.** KEGG distribution of differentially expressed gene. The ordinate is the KEGG Pathway term of enrichment, and the abscissa is the *p*-value of the term enrichment. The area of the bubbles corresponds to the number of enriched DEGs, and their color reflects the FDR as shown in the accompanying panel on the right side. (**a**) KEGG Pathway of A-vs-B; (**b**) KEGG Pathway Enrichment of A-vs-B; (**c**) KEGG Pathway of A-vs-C; (**d**) KEGG Pathway Enrichment of A-vs-C; (**e**) KEGG Pathwa of B-vs-C; and (**f**) KEGG Pathway Enrichment of B-vs-C.

### 3.6. Metabolic Pathways of Differentially Expressed Genes with Endogenous Phytohormones

Eleven DEGs were identified through pairwise comparisons among the three grades of *P. yunnanensis* seedlings. These DEGs were linked to phytohormone-related biosynthesis and signal transduction pathways (Figure 7). *AUX1* (*auxin influx carrier*, *AUX1 LAX* family) serves as a crucial auxin transporter during the biosynthesis of IAA, whose (PITA_14214) gene is down-regulated from Grade A to Grade C seedlings. However, there were two DEGs in the GH3 (*auxin responsive GH3* gene family) node that exhibited different fluctuating trends in three grades, the gene (PITA_28172) was up-regulated from Grade A to Grade C seedlings, and the GH3.6 gene (PITA_37447) was down-regulated. The results indicated that *AUX1* and *GH3* genes play major regulatory roles in the auxin signal transduction pathway and could influence plant growth. In the metabolic pathway of cytokinin signal transduction, one differential gene (PITA_28368) was down-regulated in the *AHP* (histidine-containing phosphotransfer peotein) node, and one differential gene (PITA_28306) was down-regulated in the *B-ARR* node.

Besides auxin and CTKs, DEGs associated with ABA, JA, and SA were also identified. ABA regulates stomatal closure and seed dormancy in plants. The expression of the *PP2C* (*protein phosphatase 2C*) gene showed two different fluctuation trends, which were prominently up-regulated in Grade A seedlings and gradually down-regulated from Grade B seedlings to Grade C seedlings. In the JA biosynthesis and metabolic pathways, all three DEGs related to *JAZ* signaling were up-regulated in Grade B seedlings, with lower or even no expression in Grade A and Grade C seedlings, in which the *TIYF10a* and *TIYF10b* gene families involved in the hormone signaling pathway of JA jointly regulate the ubiquitin-mediated protein hydrolysis process and seedling senescence as well as the stress response of seedlings. Moreover, one DEG was linked to the SA signaling pathway, while it not involved in the other pathways. In any case, notable differences were observed in the expression of genes involved in various phytohormone signal transduction pathways across different grades of seedlings.

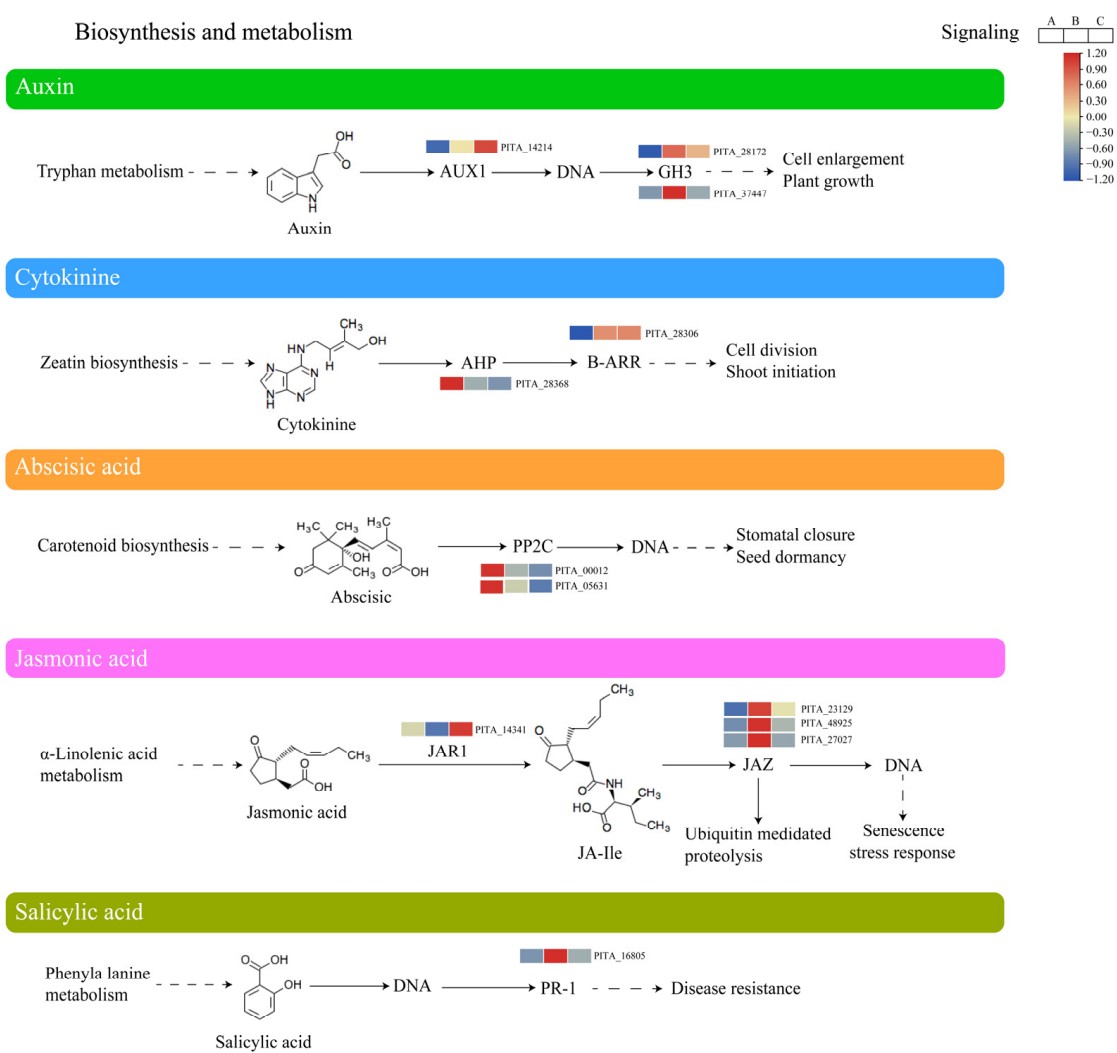

**Figure 7.** Heating maps of DEGs associated with plant hormone signal transduction in different grades of *P. yunnanensis* seedlings. From top to bottom, five panels show DEGs involved in the biosynthesis, metabolism, and signal transduction pathways of auxins, CTK, ABA, JA, and SA, respectively. Red squares mean up-regulation, and blue squares mean down-regulation.

Pearson correlation analysis revealed that (Supplementary Table S2) at the hormone content level, auxins exhibited a highly notable positive correlation with CTKs, while showing a remarkable negative correlation with ABA. Meanwhile, CTKs also demonstrated a significant negative correlation with ABA. At the hormone expression quantity level, auxins exhibited significant negative correlation with both CTKs and ABA. There was no significant difference among other hormones.

### 3.7. Validation of qRT-PCR for Phytohormone Metabolism Genes

Eight genes involved in the metabolism of auxins, cytokinin, abscisic acid, and jasmonic acid were obtained by screening the RNA-seq data of three grades of *P. yunnanensis* seedlings (Figure 8, Supplementary Table S3). Among these eight genes, there were one *AUX* (PITA_14214) and one *GH3* (PITA_28172) genes associated with auxin metabolism; one *AHP* (PITA_28368) gene related to cytokinin metabolism; two genes related to *PP2C*, a regulator of abscisic acid signaling (PITA_00012, PITA_05631); and there were one *JAR1* (PITA_14341) and two *JAZ* (PITA_23129, PITA_27027) related genes regulating jasmonic acid signal transduction. QRT-PCR validation was performed to be based on the above eight DEGs for hormone metabolism. Although there were some differences between the results obtained from qRT-PCR and RNA-seq, the expression trend was basically consis-

tent. For instance, in the transition from Grade A to C, the relative expression levels of PITA_05631, PITA_28368, and PITA_00012 showed a decreasing trend; specifically, from Grade A to B, the relative expression levels of PITA_28368 and PITA_00012 decreased by 50%, which suggested they could have more available roles in height growth of *P. yunnanensis* seedling. Furthermore, the relative expression levels of PITA_28172, PITA_27027, and PITA_14214 increased from Grade A to B and then decreased from Grade B to C; while the relative expression level of PITA_23129 was the opposite, which displayed they might have some effects on the differentiation of *P. yunnanensis* seedlings. As shown in Figure 8, only the data of PITA_23129 and PITA_14341 show different trends in RNA-Seq and qRT-PCR analysis. The expression profiles of six DEGs obtained via qRT-PCR were basically in line with those of RNA-seq, which indicated the reliability of the RNA-seq results.

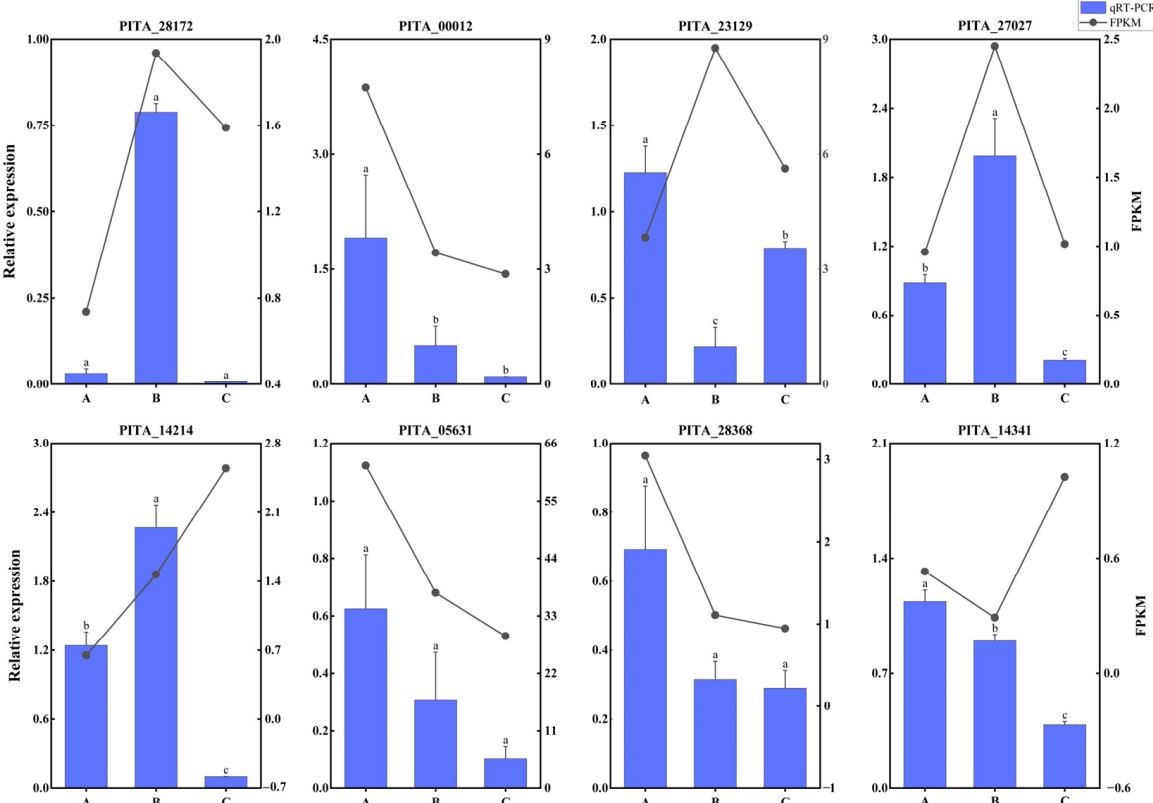

**Figure 8.** qRT-PCR validation of the expression patterns of eight DEGs in *P. yunnanensis* seedling differentiation. The *x*-axis represents the three grades; the right *y*-axis represents the FPKM values from RNA-seq; and the left *y*-axis represents the relative expression levels normalized from the qRT-PCR results. Error bars represent the variation, indicated by the standard deviation (SD), around the mean value (n = 3). Lowercase letters represent significance between different grades.

## 4. Discussion

### 4.1. Differences in the Contents and Ratios of Endogenous Phytohormones in P. yunnanensis Seedlings in Different Grades

It is generally accepted that phytohormones are trace organic compounds produced in plants, which can regulate plant growth, reproduction, resistance to both biotic and abiotic stresses, etc. Meanwhile, phytohormones are also the most important information system for transforming matter and energy into plant growth. As a signaling substance, phytohormones play a pivotal role in regulating the height growth and differentiation [46]. For example, phytohormones like GA, CTK, auxin, and JA influence plant height by regulating both cell elongation and division [47]. GA and IAA especially impact the dwarfing of bananas [48]. The decline in apical dominance in the *P. sylvestris* var. *mongolica* coincided with a remarkable reduction in IAA and CTK content contrapositive to the wild type [20].

Plant height morphogenesis is the result of continuous increase in cell number and cell size, and GA is a fundamental endogenous controller of plant height [49]. In recent decades, the functions of the phytohormone GA in determining plant height have been revealed primarily by studying mutants with GA deficiencies or excesses [50]. We also attempted to assess the levels of GAs, which have been investigated in a prior study, but no GA was detected in the current study. It is possible that genes were expressed but no metabolites were detected [11]. Increasing evidence suggests that phytohormones interact synergistically with one another [51]. Wang et al. demonstrated that ABA negatively impacts plant growth, resulting in dwarfing, which is often associated with GA antagonism [52]. Zhao et al. regulated plant height in rice by modulating the GA and ABA metabolisms, and they found through their research results that ABA and GA had an antagonistic effect [51]. However, Zeevaart et al. found that the mutants lacking ABA showed dwarf phenotype and believed that endogenous ABA played a role in promoting plant growth [53]. The results of this study were similar, suggesting that ABA in different grades of *P. yunnanensis* may be involved in promoting plant height growth, and its physiological and molecular mechanisms need to be further investigated. In addition, the IAA content of Grade B seedlings was higher than Grade C and A seedlings (Figure 2c). Research indicated that auxins have an extreme impact on the early growth, development, and morphogenesis of plants [54,55]. It can coordinate with cytokinins and other phytohormones to modulate seedling differentiation growth. Ethylene biosynthesis can be induced by regulating the activity of ACC synthase, a pivotal enzyme in this process [56]. On the other hand, JA (jasmonic acid) and SA (salicylic acid) are generally considered to be stress hormones [57]. In this investigation, the levels of JA and SA were elevated in *P. yunnanensis*, which probably regulated the resistance of *P. yunnanensis* wood nematode (Figure 2c). It is generally believed that auxins and cytokinins are hormones that promote plant growth and development, while ABA is a stress-response phytohormone [57]. The ratios of auxins/ABA, CTKs/ABA, and (auxins + CTKs)/ABA were significantly increased in Grade B and C seedlings than those in Grade A seedlings. Plant height may be controlled not only by the phytohormone content but also by the ratio of endogenous phytohormones, which can reflect the dominant roles occupied by growth-promoting and growth-inhibiting phytohormones [58]. These results indicated that the content of endogenous phytohormones and their balance might be associated with the high growth of *P. yunnanensis* seedling. The outcomes of this research is similar to the results of the comprehensive analysis of shoot culm and the study of auxin-related genes of *Phyllostachys edulis* by Zhang et al. [59]. Physiological processes during plant growth, like seedling hypocotyl elongation and plant height growth, are jointly regulated by a series of phytohormones, such as auxin, cytokinin, etc., which also reveals the intricate interactions and feedback regulation among different plant hormones [60].

### 4.2. Analysis of DEGs of P. yunnanensis among Different Grades

Plant growth is affected by its own genetic material, and through the interaction of gene expression products, an intricate regulatory network involving numerous genes is formed through multiple biological and metabolic pathways to regulate plant growth [61,62]. In the present research, we selected *P. yunnanensis* seedlings from three different grades that displayed inherent variations in plant height. This selection aided the exploration of the molecular mechanisms underlying the variations in plant height among *P. yunnanensis* seedling. Transcriptome analysis has enabled the identification of novel genes, and the expression patterns of these genes could offer valuable insights on plant height. Additionally, differential expression analyses indicated that the most DEGs were present in cell composition and molecular function in different grades of *P. yunnanensis* seedlings. The most functional cells in *P. yunnanensis* are relatively mature, and certain differential genes may represent transcription factors responsible for maintaining the morphological differences among the three grades of seedlings. Subsequently, KEGG enrichment analysis showed that phytohormone signaling, endocytosis, and plant–pathogen interaction were the three primary pathways commonly shared among all three groups (Figure 6a–c).

Research has demonstrated that plant hormone signal transduction was linked to plant height, and inhibiting plant hormone metabolism and signal transduction pathways is one of the mechanisms for seedling differentiation [63,64]. Zang et al. proposed that the dwarf phenotype was caused by the disrupted plant hormone signaling, leading to a reduced rate of stem growth and the decreased accumulation of lignin [65]. Sheng et al. indicated that plant hormone signaling is among the key mechanisms influencing plant dwarfing [50]. The plant hormone signal transduction pathway was responsible for regulating the expression of plant hormones to control cell growth, and various physiological activities of metabolic pathways provided energy for cell activities [46]. This indicates that the key differential genes involved in phytohormone signal transduction pathway may be closely associated with the seedling height.

### 4.3. Relationship between Plant Growth-Related Hormone Signal Transduction Gene Expression and Plant Height Growth

In recent years, studies on apical dominance formation have predominantly concentrated on phytohormone synthesis, transport, signal transduction, and metabolism [12]. In *Populus tomentosa*, significant variations were observed in genes related to GA, ABA, ETH, IAA, CTK, and JA pathways. This implies that multiple phytohormones play roles in inducing endodormancy and governing the developmental processes after the induction of dormancy [66–68]. The process of seedling height establishment is closely linked to cell division and differentiation, with the regulation of phytohormone signaling playing a crucial role, primarily by controlling cell elongation factors and reconstructing the cell wall polysaccharide network [69,70].

The auxin signaling transduction is achieved through the expression of genes (*AUX/IAA* proteins and the *GH3* gene family, etc.), regulating plant growth and development [61]. Alterations in the auxin pathway may impede stem elongation, thus affecting plant height [71,72]. *AUX1* and *GH3*, as early-response factors to auxin, can rapidly react to auxin induction, thereby controlling cell elongation and plant growth [73]. Previous studies indicate that the *GH3* gene family is associated with plant growth, regulating auxin balance. In *Arabidopsis*, the *GH3* gene can alter auxin concentrations, thereby controlling root and hypocotyl growth, with the overexpression of this gene resulting in stunted growth in adult plants [74]. Lan Feng et al. conducted a study on the castor bean *ARF*, *GH3*, and *AUX/IAA* gene families, involving the whole-genome identification and analysis of gene expression patterns in tall and dwarf varieties [75]. Their research revealed that *AUX/IAA* proteins and genes associated with auxin within the *GH3* gene family are involved in determining the plant height of tall and dwarf castor bean varieties. The findings of this study are consistent with their research.

Existing research suggests that the cytokinin signaling transduction system is analogous to bacterial two-component response systems [76]. In the plant genome, genes similar to bacterial two-component signaling elements have been discovered, such as histidine kinases, histidine phosphotransfer proteins, and response regulators [77]. Histidine phosphotransfer proteins (*AHP*) serve as positive regulatory proteins in the cytokinin signaling transduction pathway [78]. They have the ability to activate *B-ARR*, which in turn activates the gene transcription of *A-ARR*, ultimately influencing cell division and apical bud sprouting [79]. This study has identified that *AHP* and *B-ARR* may be up-regulated during the early response factor initiation, but significantly down-regulated as the downstream response progresses (Figure 7). In grade A seedlings, there was a notable up-regulation in the expression of cytokinin-related gene *AHP*, indicating that grade A seedlings undergo more frequent cell division activities during the fast-growth period compared to grade B and C seedlings, thereby confirming the involvement of *AHP* in cytokinin signal transduction (Figure 6). Suzuki et al. demonstrated that providing evidence for the involvement of *AHP* in cytokinin signal transduction [80]. Qi et al. demonstrated in the molecular mechanism of cytokinin signal transduction that *B-ARR* is involved in cytokinin signal transduction and can indue *A-ARR* expression [79].

In this study, we observed the differential expressions of *TIFY* and *PP2C* genes involved in ABA and JA signal transduction pathways. Wang et al. identified the *TIFY* gene family in roses and found that its gene members play a role in the wide-ranging regulatory systems within plant cells, participating in the regulation of various aspects of plant life [81]. Additionally, this gene family regulates the expression of *JAZ* family genes by ABA, indicating that the effect of endogenous phytohormones on plant growth is not solely determined by the individual hormone levels but rather by the synergistic effects of multiple hormones working in concert to comprehensively coordinate seedling growth and development (Figure 7). Furthermore, phytohormone regulation is influenced by the relevant gene expressions in signal transduction pathways, and seedling growth is subject to bidirectional feedback regulation between hormone-related genes and the external environment [60]. Dong et al. through their research on the heterologous expression of potato *PP2C* in *Arabidopsis* demonstrated that there is a mutual signaling interaction between auxins and ABA to regulate plant apical dominance. At the same time, the *PP2C* genes may be associated with plant drought resistance [82]. Wang et al. conducted verification studies on *PP2C* in Arabidopsis, confirming its regulatory function in ABA signaling transduction pathway [83]. Furthermore, the outcomes of stress treatments suggested that *PP2C* genes are involved in stress response mechanisms. Hence, it is hypothesized that the functions of the genes PITA_05631 and PITA_00012 within the *PP2C* gene family in *P. yunnanensis* seedlings may be associated with stress responses, resistance mechanisms, and the modulation of stomatal conductance (Figure 7). By modulating abscisic acid hormone levels, these genes could enhance seedling resilience and adaptability to the environment. This, in turn, could improve stomatal conductance, enhance water use efficiency, and optimize photosynthesis, potentially boosting the competitive advantage of *P. yunnanensis* seedlings for better growth and development.

## 5. Conclusions

To sum up, this study is aimed to investigate the underlying mechanisms of *P. yunnanensis* seedling differentiation using phenotypic, transcriptomic, and endogenous phytohormone analyses. It provides a comprehensive knowledge of seedling differentiation in *P. yunnanensis*. The results of contents and ratios of phytohormones suggested that individual phytohormones have a limited capacity to regulate gene expression, and seedling differentiation is caused by phytohormones modulation. However, the imbalance in hormone ratios may be a primary factor contributing to the variations in height among *P. yunnanensis* seedlings. Transcriptomic analysis of *P. yunnanensis* identified DEGs involved in phytohormone signal transduction pathways, plant–pathogen interaction, etc. In addition, several candidate genes associated with phytohormone biosynthesis and signal transduction pathways were identified, including *AUX1*, *GH3*, *AHP*, *B-ARR*, *PP2C*, etc. which provided clues for future research. These findings provide insights into the molecular genetic control of seedling height growth of *P. yunnanensis*.

**Supplementary Materials:** The following supporting information can be downloaded at: https://www.mdpi.com/article/10.3390/f15030489/s1, Supplementary Table S1. Growth status of *P. yunnanensis* seedlings of different grades in different months. Supplementary Table S2. Pearson correlation between plant height and hormone content and expression quantity of different grades of *P. yunnanensis*. Supplementary Table S3. Primers of differentially expressed genes qRT-PCR.

**Author Contributions:** Conceptualization, Y.X., N.C. and L.C.; methodology, Z.L. and Y.X.; investigation, Z.L., Q.W. and Z.Y.; writing—original draft preparation, Z.L., Q.W. and Z.Y.; writing—review and editing, Y.X., N.C. and L.C.; funding acquisition, Y.X. All authors have read and agreed to the published version of the manuscript.

**Funding:** This research was funded by National Nature Science Foundation of China (NSFC), grant number 32360381; Key Project of Joint Funds of the Basic Agricultural Research of Yunnan Province, grant number 202301BD070001-152; Yunnan Young & Elite Talents Projects, grant number YNWR-QNBJ-2019-075; and Joint Funds of the Basic Agricultural Research of Yunnan Province, grant number 202301BD070001-035.

**Data Availability Statement:** Raw sequences have been deposited in the Sequence Read Archive under Bioproject PRJNA1065384.

**Acknowledgments:** Thank you very much for the help of the authors and the support of the research group.

**Conflicts of Interest:** The authors declare no conflicts of interest.

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
