# Peer review of "Endogenous Phytohormone and Transcriptome Analysis Provided Insights into Seedling Height Growth of Pinus yunnanensis"

_forests, doi:10.3390/f15030489_

Round 1

Reviewer 1 Report

Comments and Suggestions for Authors

Manuscript ID: forests-2854746

Title:  Endogenous hormone and transcriptome analysis provided insights into Pinus yunnanensis seedling height growth

Authors: Lu et al.

General comments:

The manuscript deals with the analysis of three different stages/grades of ‘Pinus yunnanensis seedlings and tries to decipher the molecular basis of the observed differences in the growth pattern. The major focus is on transcriptomic analysis, phytohormone content and their ratios, key important genes involved in phytohormone synthesis, which over all concludes that the Pinus yunnanensis seedling growth pattern is outcome of involvement of multiple phytohormones. Overall this is a very broad statement and the study may need experimental support for certain statements. The current version of the manuscript in certain sections such as discussion, is too much descriptive and lack focus, which in other it tries to make concluding statements with relatively less data. Statistical analysis, particularly for qRT-PCR is not evident. It is also important for validation of contrasting trend in RNA-Seq and qRT-PCR data. Hence, many sections of the manuscript may need to be improved.  Some of the key concerns are listed below and also indicate in the PDF file of the manuscript.

Section specific comments:

Title:                    The title of the manuscript is appropriate, however it suggested to use ‘phytohormone’ in place of ‘hormone’.

Abstract:             Abstract looks very long and may be minimized a little bit. Rationale of the study in view of the known information on plant/seedling growth must be highlighted. Better to use ‘phytohormone’ in terms of ‘hormones’ in Abstract and other sections of the manuscript. Some suggestions are indicated in the PDF file for improvement, while few are listed below:

Line 17:               ‘Significantly distant’ is not clear. It is in comparison to what? This statement may need to be rewritten for clarity.

Lines 32-33:        May be rewritten for enhanced clarity.

Lines 41-42:        Were found means what? is not clear here. May be rewritten.

Introduction:     The length of the introduction section is appropriate and written in a fine manner. Some minor changes have been suggested (see PDF file for details) and few are briefly mentioned here:

Lines 56-58:        Statement may be re-written for better clarity.

Lines 64-67:        Add references in support?

Line 81:               Plant height is reduced due to reduced internode length. The statement may be modified accordingly, if needed.

Line 107:             Why these two contrasting traits, 'drought resistance' and poor drought tolerance' have been mentioned for the same plant species. It must be cross checked and rectified?

Line 125:             This statement lacks clarity? Does this indicate differences between different sub-species of P. yunnanensis?

Lines 136-137:   This statement seems disconnected and may be revised w.r.t. the focus of the study.

Material and methods: This section may need to be improved. See the PDF file for comments and some points below:

Line 152:             What data was measured to decided and select particular types of seedlings for analysis?

Lines 157-158:    This statement lacks clarity, may be rewritten.

Lines 162-164:    Seems repetitive like previous description, may be removed or minimized?

Line 167:             How nanodrop was used for analysis of integrity of RNA preparation? Generally, other approaches are used for this aspect?

Lines 169-170:   This should be moved earlier and combined with the similar description on RNA assessment above.

Lines 172-173:   What were the primers used for RT reaction once mRNA was fragmented, is not clear?

Line 174:             Only DNA polymerase, or some other components were also added?

Lines 179-183:   Add some Refs here?

Lines 189-190:   Kindly check if "GO analysis is used to identify false genes"?

Lines 193-194:   This statement is also not very clear, it may be rewritten.

Lines 205-213:   If this is a standard way of calculations, it may not be needed in detail. Reference may be provided, instead.

Lines 215-217:   This Kit is only for isolation, and not for Reverse Transcription. Kindly re-check and rectify.

Lines 218-222:   How are the primers designed for qRT-PCR? and what are their characteristics?

Lines 234-236:   How significance of qRT-PCR data was evaluated is not mentioned?

Results:               Results section will need improvement at several places. In particular, the content/description should be written in a manner to avoid unnecessary lengthy details and content which is also evident in the figure/tables. See comments/suggestions in the PDF file and also below:

Line 239:             This may be part of introduction/discussion section.

Lines 242-243:   This is not very clear, as the data is simple plot o plant height vs time. What type of data fitting was done here may be elaborated.

Line 246:             This is was anyway the seedling were selected in the beginning of the experiment.

Lines 249-250:   How this analysis was done for assessment of growth rates, and what statistical analysis was done for significant differences?

Lines 251-252:   Significance or not significant may be supported by analysis? It must be included in the figure/table data.

Line 254:             A short seedling is likely to remain relatively short that a normal seedling, as both will get the same duration and seasonal parameters till full maturity. Can authors explain why the seedling specific curves will not behave as they are visible in the Figure 1?

Line 276:             C > A = B, seems to be correct as per graphs, as levels in A and B are not statistically different.

Line 277:             Re-write the description to indicate 'lower or higher' values.

Line 282:             Which other hormones are indicated here?

Lines 293-295:   This statement is the novel finding of this study or it is known that Phytohormones do interplay in plant development at almost all stages?

Line 307:             Which physiological parameters are indicated here?

Line 315:             In manuscript, Figure Table 1 should be placed before Figure 3.

Lines 318-319:   This statement lacks clarity, may be re-written.

Line 342:             The clustering is done here based on the Expression levels. How it will discover the biological relationships?

Lines 343-347:   Long statement, difficult to follow, may be split into small statements.

Line 349:             Explain the color coding and scale indicating the level of expression.

Lines 358-360:   This may be removed from the results section. If needed add briefly into M&M section.

Lines 373-375:   'Catalytic activity pathway' seems not so commonly used. The term indicated collectively the proteins involved in enzyme activities. It may be be referred to a pathway. This may be clarified.

Lines 386-405:   Minor changes suggested for improvement.

Lines 406-407:   This statement may need further experimental evidences, other than just differential levels.

Lines 417-419:   Some of this content is appropriate for introduction/discussion.

Lines 422-429:   This description may be minimized a little bit.

Lines 431-435:   Certain amount of content appropriate for discussion.

Lines 451-457:   Is this type of relationship among different phytohormone levels in plants generally reported?

Lines 471-473:   The M&M indicated qRT-PCR based on double -delta Ct method, which gives a fold-change. However results show relative expression. Kindly clarify in M7M how it was done.

Lines 479-481:   This conclusion may need some more experiments than just relative levels.

Line 482:             The data of PITA 23129 and PITA 14341 show inverse trend in RNA-Seq and qRT-PCR analysis? Which data is more believable to explain the involvement in the observed results? Also the statistical analysis of qRT-PCR is not evident. It may be included for meaningful conclusions.

Line 484:             Statistical analysis of qRT-PCR is not evident. It may be included for meaningful conclusions.

Discussion:         The discussion section is very lengthy and contains, several statements similar to results, and references of publications which may have less relevance to the present study. This may be minimized and this section can be improved.

Lines 491-492:   May need to be re-written to enhance clarity.

Lines 497-500:   More like results section, may be removed/minimized.

Lines 504-506:   Can this be verified by any other means? Or if there are similar reports showing expression of GA genes with no metabolite detection, may be cited.

Lines 522-524:   There was no external treatment of Phytohormones in this study.

Lines 526-528:   More like results section, may be removed/minimized.

Lines 536-537:   This statement or similar statement is present at multiple places, this may be avoided.

Lines 538-542:   This may not suitable for the theme of this manuscript. If needed, it may be minimized.

Lines 551-557:   More like results section, may be removed/minimized.

Lines 560-564:   More like results section, may be removed/minimized.

Lines 565-582:   This description may be written in a concise manner in view of result and analysis of the present study.

Lines 600-641:   Description may be minimized by writing in a concise manner, removing non-relevant discussion from previous reports, and by removing te content similar to results.

Lines 643-645:   More like results section, may be removed/minimized.

Line 669-675:     This concluding paragraph may be merged with the next section '5- Conclusions'

Conclusions:      This section may be minimized by avoiding repetitive content as well as the suggestions/comments in the previous sections.

Figures:               The legends may be made more informative

Comments on the Quality of English Language English may need minor corrections at certain places

Author Response

Dear Editor and reviewers,

Thank you for your letter and for the reviewers’ comments concerning our manuscript entitled “Endogenous phytohormone and transcriptome analysis provided insights into Pinus yunnanensis seedling height growth”. After the reviewing process, reviewers have suggested major revisions to this manuscript. We would like to thank all the suggestions and corrections. We really appreciate the great effort made by reviewers and editors which undoubtedly has improved at a great extent the quality of the manuscript. We have agreed to include all the changes proposed by the reviewers to text, tables and figures to improve the manuscript and make it more clear and readable.

In order to facilitate the reviewing process, a brief response to the queries or comments made by reviewers is given in the following point by point reply. Reviewer comments are in black and our responses in red. The revised parts are marked up by using the "Track Changes" function in the attached manuscript. I hope the text reaches now the level for publication in Forests.

Reviewer 2 Report

Comments and Suggestions for Authors

Comments on the Quality of English Language

I find the sentence construction in this text to be confusing, which hinders its fluidity and clarity. I would recommend a thorough revision of the English to improve comprehension and clarity. Discussion in particular needs a lot of improvement.

Author Response

(The authors gave the same response as above.)

Round 2

Reviewer 1 Report

Comments and Suggestions for Authors

Manuscript ID: forests-2854746-Revised Version

Title:  Endogenous phytohormone and transcriptome analysis provided insights into Pinus yunnanensis seedling height growth

Authors: Lu et al.

General comments:

The revised manuscript has been considerably improved. Kindly re-assesses the qRT-PCR statistical data of two panels: PITA_00012 and PITA_28368 for significance. See the language and style of writing throughout.

Comments on the Quality of English Language

Quality of English language is fine.

Author Response

Thank you for your letter and for the reviewers’ comments concerning our manuscript entitled “Endogenous phytohormone and transcriptome analysis provided insights into Pinus yunnanensis seedling height growth”. After the reviewing process, reviewers have suggested revisions to this manuscript. We would like to thank all the suggestions and corrections. We really appreciate the great effort made by reviewers and editors which undoubtedly has improved at a great extent the quality of the manuscript. We have agreed to include all the changes proposed by the reviewers to text, tables and figures to improve the manuscript and make it more clear and readable.

In order to facilitate the reviewing process, a brief response to the queries or comments made by reviewers is given in the following point by point reply. Reviewer comments are in black and our responses in red. The revised parts are marked up by using the "Track Changes" function in the attached manuscript. I hope the text reaches now the level for publication in Forests.
